# Nutrient Sensing and Biofilm Modulation: The Example of L-arginine in *Pseudomonas*

**DOI:** 10.3390/ijms23084386

**Published:** 2022-04-15

**Authors:** Chiara Scribani Rossi, Laura Barrientos-Moreno, Alessio Paone, Francesca Cutruzzolà, Alessandro Paiardini, Manuel Espinosa-Urgel, Serena Rinaldo

**Affiliations:** 1Laboratory Affiliated to Istituto Pasteur Italia, Fondazione Cenci Bolognetti-Department of Biochemical Sciences “A. Rossi Fanelli”, Sapienza University of Rome, 00185 Rome, Italy; chiara.scribanirossi@uniroma1.it (C.S.R.); alessio.paone@uniroma1.it (A.P.); francesca.cutruzzola@uniroma1.it (F.C.); alessandro.paiardini@uniroma1.it (A.P.); 2National Biofilms Innovation Centre, Biodiscovery Institute-School of Life Sciences, University of Nottingham, Nottingham NG7 2RD, UK; laura.barrientos@eez.csic.es; 3Department of Biotechnology and Environmental Protection, Estación Experimental del Zaidin, CSIC, 18008 Granada, Spain

**Keywords:** arginine, biofilm, *Pseudomonas*, nutrients, c-di-GMP, ArgR, RmcA, metabolism

## Abstract

Bacterial biofilm represents a multicellular community embedded within an extracellular matrix attached to a surface. This lifestyle confers to bacterial cells protection against hostile environments, such as antibiotic treatment and host immune response in case of infections. The *Pseudomonas* genus is characterised by species producing strong biofilms difficult to be eradicated and by an extraordinary metabolic versatility which may support energy and carbon/nitrogen assimilation under multiple environmental conditions. Nutrient availability can be perceived by a *Pseudomonas* biofilm which, in turn, readapts its metabolism to finally tune its own formation and dispersion. A growing number of papers is now focusing on the mechanism of nutrient perception as a possible strategy to weaken the biofilm barrier by environmental cues. One of the most important nutrients is amino acid L-arginine, a crucial metabolite sustaining bacterial growth both as a carbon and a nitrogen source. Under low-oxygen conditions, L-arginine may also serve for ATP production, thus allowing bacteria to survive in anaerobic environments. L-arginine has been associated with biofilms, virulence, and antibiotic resistance. L-arginine is also a key precursor of regulatory molecules such as polyamines, whose involvement in biofilm homeostasis is reported. Given the biomedical and biotechnological relevance of biofilm control, the state of the art on the effects mediated by the L-arginine nutrient on biofilm modulation is presented, with a special focus on the *Pseudomonas* biofilm. Possible biotechnological and biomedical applications are also discussed.

## 1. Introduction

Although bacteria can be found as free-living individuals, biofilms are considered their prevailing mode of life in the environment [1,2]. These are multicellular communities, usually bound to a solid surface, in which microorganisms live embedded in a self-produced extracellular matrix with structural and protective roles [3]. Biofilm development is a widespread trait in prokaryotes that was probably acquired early during their evolution [4]. It is a key strategy for microorganisms to colonise, adapt and survive in very different and changing habitats for several reasons. Firstly, bacteria establish social interactions within the biofilm, such as competition or metabolic cooperativity, where they can exchange, remove and redistribute nutrients [2]. Secondly, biofilms constitute a favourable environment to attain new traits through horizontal gene transfer, which can confer an adaptive advantage by increasing genetic diversity and accelerating population evolution [5]. Finally, the extracellular matrix provides protection against a variety of environmental challenges, from dehydration to antimicrobial compounds [6].

Much of our current knowledge on the molecular mechanisms involved in the process of biofilm development comes from studies in *Pseudomonas* species, particularly of opportunistic pathogen *P. aeruginosa*. Planktonic and surface-attached cells show significant differences in terms of global gene expression and protein profile. As an example, a proteomic study in *P. aeruginosa* revealed clear modifications between those two lifestyles, including proteins involved in signal transduction, second messenger systems, appendages and outer membrane components, among others [7]. Gene expression changes have been observed at different stages of biofilm development, or even at the same stage depending on the surrounding conditions and the spatial distribution of cells within the biofilm [8,9]. Although the regulatory network involved in biofilm formation is very complex and far from being fully understood, showing variations from one species to another, it is possible to establish three broad levels of modulation of the process by different types of signalling mechanisms.

The first level is due to environmental cues. These include surface characteristics like charge or roughness and external factors like pH, temperature or nutrient availability. The importance of nutrients in biofilm development is discussed in more detail in the following section. Other compounds and microelements like calcium, magnesium, inorganic phosphate or iron have been reported to influence attachment of *Pseudomonas* to solid surfaces [10,11,12]. In many cases, detection of and response to a signal are carried out by two-component systems formed by a sensor histidine kinase which senses the environmental stimuli and by a response regulator that transduces the signal inside the cell.

The second level of regulation comes from intracellular small molecules known as the second messengers. Usually, these are cyclic nucleotides, mono- or dinucleotides, which are widely distributed in both prokaryotes and eukaryotes. In terms of biofilm formation and transition between lifestyles, cyclic di-guanosine monophosphate (c-di-GMP) has emerged as a key element in many bacteria. Variations in local and global cellular contents of this molecule modulate different functions, including motility, secondary metabolism, or surface attachment [13,14]. As a rule, high cellular levels of c-di-GMP correlate with the sessile lifestyle, and low levels promote motility. Production of the structural elements of the biofilm matrix (adhesins, exopolysaccharides and possibly also extracellular DNA) is controlled by c-di-GMP in many microorganisms, including *P. aeruginosa* [15], *P. fluorescens* [11] and *P. putida* [16].

Finally, the third level of modulation is derived from intercellular communication via quorum sensing (QS). QS is based on small diffusible molecules called autoinducers produced by bacteria and released to the environment at a low basal rate. When a critical threshold concentration of these molecules in the local environment is reached, QS mechanisms are activated, leading to coordinated expression of specific genes in a cell density-dependent manner [17]. This process allows bacteria to synchronise particular behaviors and thus act like a multicellular organism, as well as to control a wide range of functions, some of which influence biofilm development [18].

Environmental, intercellular and intracellular signals do not lead to independent regulatory cascades but are interconnected, resulting in a complex network that determines the expression and activity of transcriptional and post-transcriptional regulators controlling different aspects of biofilm formation and dispersal. In the following sections, we focus on the role of nutrient signals, and most specifically amino acid L-arginine, in the sessile lifestyle of *Pseudomonas* by presenting the state of the art on its relevance as a nutrient, a biofilm modulator and an ATP source.

## 2. Importance of Nutrients in *Pseudomonas* Biofilms

Bacteria can reshape their metabolism in response to the availability of carbon sources, nitrogen sources (both inorganic or organic, including amino acids, depending on the cellular background), ions and essential elements.

The influence of nutrients and media composition was already recognised in early mechanistic studies of *Pseudomonas* biofilm development. In *P. fluorescens*, for example, it was observed that different carbon sources (glucose, citrate or glutamate) stimulated biofilm formation through partly overlapping pathways [10]. In *P. putida*, nutrient exhaustion leads to rapid biofilm dispersal, a process that indirectly involves the turnover of the large adhesin LapA [19]. The stringent response, mediated by (p)ppGpp, has been shown to participate in starvation-induced dispersal of biofilms by positively regulating phosphodiesterase BifA and negatively regulating LapA and its transport system [20]. In *P. aeruginosa*, nutrient availability determines not only biofilm formation and dispersal, but also the biofilm structure, as predicted by in silico models and further corroborated in vivo [21]. Although there is still limited knowledge regarding the effects of specific nutrients, their importance reflects the fact that biofilm formation is an active, energy-demanding process, as well as a means for bacteria to occupy favourable, nutrient-rich environments.

The molecular mechanisms by which different nutrients influence biofilm formation or stability are being explored from various perspectives, from signal transduction through chemoreceptors [22] to population dynamics and evolutionary aspects [23]. In a recent study using *P. fluorescens* Pf01 as a model organism, the impact of a wide array of nutrients on biofilm formation was analysed, using glycerol as a carbon source to define the effect of compounds that may not be strictly “nutrients” for this microorganism but still act as environmental signals [24]. This work allowed identifying organic molecules with an effect on expression or functionality of enzymes involved in c-di-GMP turnover, thus modulating biofilm formation. Relevant molecules include specific sugars, organic acids and amino acids. Among these, L-arginine stands out as a key element based on growing evidence not only in *Pseudomonas*, but also in many other bacterial species.

## 3. L-arginine as a Pleiotropic Nutrient

L-arginine is one of the most versatile nutrients leading to multiple effects on cell metabolism both in eukaryotes and prokaryotes. Beyond its role as a building block of proteins, this amino acid may fuel metabolism acting as both a carbon/nitrogen source [25] and an ATP source under specific environmental conditions [26]. Since L-arginine is at the crossroads of different metabolic pathways involving dedicated enzymes which in principle may compete for the same metabolite [27], it has been proposed that multiple intracellular L-arginine pools may accumulate, which are differentially accessible to various metabolic enzymes [28]. Interestingly, this differential metabolite accumulation has also been proposed and demonstrated for the second messenger c-di-GMP, whose regulation may involve differentiated local c-di-GMP pools within the cell [29].

We briefly present the metabolic and genetic organisation of L-arginine metabolism found in *Pseudomonas aeruginosa* (Figure 1), which is the reference system for biofilm studies, considering that the complex metabolic network involving this amino acid could be different in diverse organisms [30]. Overall, the pathways seem to be conserved for the most part in other *Pseudomonas*, based on available in silico information (https://www.genome.jp/kegg/ (accessed on 23 March 2022)).

L-arginine biosynthesis proceeds from L-glutamate via ornithine; drawing on the carbamoyl phosphate pool, ornithine is converted into L-arginine, and the expression of the enzymes catalyzing these final steps are sensitive to L-arginine depletion [31].

L-arginine catabolism, on the other hand, includes four different routes which touch different aspects of cellular metabolism. The main route is the arginine succinyltransferase pathway (AST), which sustains the succinate/L-glutamate pool and the ammonia pool (carbon, nitrogen and energy). This metabolism is relevant under aerobic conditions, and the corresponding enzymes are encoded by the *aru* operon, which is L-arginine-inducible together with the glutamate dehydrogenase isoform encoded by the *gdhB* gene. L-arginine degradation is controlled in *P. aeruginosa* by two-component response regulator CbrA/CbrB [32], whose cascade induces the expression of transcriptional activator ArgR responsible for L-arginine biosynthesis repression and L-arginine catabolism activation [33]. 

The other two catabolic routes, less represented, are the L-arginine decarboxylase (ADC) and dehydrogenase (ADH) pathways, the latter re-evaluated as the aminotransferase pathway (ATA) [31,34]. These are controlled by other metabolites, particularly polyamines, rather than L-arginine.

A fourth peculiar pathway is the arginine deiminase (ADI) one, which is the most relevant for L-arginine utilization under anaerobic conditions, given that it allows *P. aeruginosa* to sustain ATP production from carbamoyl phosphate in place of the oxygen-dependent respiratory chain [35]. The ADI pathway, encoded by the *arc* operon, is mainly regulated by the ANR transcription factor [36], ArgR being a helper to augment the transcription activation [37]; ANR is a global regulator of anaerobic metabolism, and is also responsible for the induction of the nitric oxide (NO)-sensitive DNR transcription factor [38]. This second regulator is a heme-based sensor able to bind to NO (nitrate-derived) and, in turn, to transcriptionally activate the denitrification pathway, the anaerobic respiratory chain where nitrate is the final electron acceptor to be sequentially reduced to dinitrogen [39,40]. Denitrification induction leads to a nitrate-dependent (partial) repression of the ADI pathway to favour the more efficient respiratory process instead of the fermentative one [41]. The relevance and functionality of the ADI pathway in other *Pseudomonas* remains to be fully characterised, particularly in those species considered to be strict aerobes, such as *P. putida*.

From a metabolic point of view, L-arginine catabolism allows this amino acid to act as a key precursor of many other nutrients and modulators, including in other organisms; urea, creatine and L-glutamate homeostasis are directly linked to L-arginine degradation, while its biosynthesis may sustain L-glutamate and pyrimidine biosynthesis by also sharing the regulatory strategy [42].

Among the modulating molecules, L-arginine is responsible for the synthesis of polyamines, a group of (multi)aminic metabolites able to affect cell division and proliferation by controlling gene expression, DNA replication and protein biosynthesis [43]. Arginine and polyamines have also been described to modulate siderophore-mediated iron acquisition and tolerance to oxidative stress in *P. putida* [44]. 

Moreover, in the late 1980′s, it was discovered that the endothelium-derived relaxing factor, a signal molecule crucial in the mammals’ vascular biology, corresponds to NO [45] and originates from L-arginine thanks to the nitric oxide synthase activity (NOS) [46,47]. L-arginine-derived NO is one of the strategies of the host immune system to tackle bacterial infections; for this reason, L-arginine supplementation may represent a therapeutic approach for cardiovascular disorders [48], wound healing [49] and immunostimulation [50]. Nevertheless, it may represent a Janus molecule in cancer therapy, being both an immunonutrient and an onconutrient [51,52]. In this way, pathogens such as *P. aeruginosa*, compete with immune cells for nutrients (see below for details).

Although the large part of NO in bacteria originates from inorganic sources including nitrates and nitrites [53], a bacterial counterpart of eukaryotic NOS, bNOS, has been reported in certain species; bNOS-derived NO can play different roles, including toxin biosynthesis, protection against oxidative stress, a growth factor in response to stress and an environmental signal to trigger the target tissue growth [54]. 

Given the large repertoire of metabolic pathways involving L-arginine, their study has been conducted to improve the industrial production of L-arginine (and its derivatives) by engineered overproducing strains [34,55]. Moreover, engineering of proteins involved in L-arginine recognition has been useful to develop a biosensor for L-arginine detection in biological fluids, up to now possible through HPLC and derivatization [56].

L-arginine is one of the few nutrients used to tackle oral biofilms, which also includes *P. aeruginosa* [57], as a strategy to target pathogens without destroying the physiologically present microbial flora, which contributes to controlling the development of oral infectious diseases [58]. The current approach aims at defining nutrient-based solutions to antagonise biofilm virulence by preventing microbial dysbiosis. 

## 4. L-arginine Sensing and Biofilm Regulation in *Pseudomonas*

As mentioned previously, certain amino acids are emerging as key molecules in the regulation of attachment and biofilm development in different microorganisms. In *P. aeruginosa*, several amino acids promote robust biofilm formation and reduce swarming motility: arginine, ornithine, isoleucine, leucine, valine, phenylalanine and tyrosine [59]. In *P. putida*, arginine and its precursor aspartic acid have divergent effects on biofilm formation; while increasing concentrations of arginine stimulate surface attachment, aspartic acid causes the opposite effect. These phenotypes relate to changes in the c-di-GMP content [60]. Other amino acids also influence accumulation of the second messenger in this bacterium, with proline and valine reducing it and tryptophan causing an increase [44]. Interestingly, there is a synergistic effect between arginine and tryptophan [44]. All these data suggest the existence of complex signalling circuits that connect amino acid sensing, metabolic pathways and the cellular mechanisms controlling bacterial lifestyles. Some of those circuits are being uncovered.

Once the diverse environmental cues are perceived, the intracellular levels of c-di-GMP change to reshape cellular metabolism; biosynthesis and degradation of this dinucleotide are due to the activity of diguanylate cyclases and phosphodiesterases, harboring the conserved GGDEF and EAL domains, respectively [14]. In *P. aeruginosa*, L-arginine supplementation leads to increased intracellular levels of c-di-GMP through the activity of SadC and RoeA diguanylate cyclases; the corresponding mutant strains showed a reduction in biofilm formation when grown with this amino acid [59].

Among the GGDEF-EAL-containing proteins, there are multidomain enzymes harboring putative nutrient-sensory domains, thus suggesting that selected nutrients may tune the c-di-GMP network by controlling enzyme activity [24]. Indeed, a sensory system of L-arginine able to modulate the c-di-GMP levels has been indirectly identified in *Salmonella typhimurium* [61] or hypothesised in other organisms [60,62]. It has also been proposed that cellular arginine pools, resulting both from biosynthesis and uptake of the amino acid from the environment, are sensed and transduced into changes in c-di-GMP contents [44]. This hypothesis is based on the fact that mutants defective in either arginine synthesis or in its transport show reduced second messenger levels. This would imply a connection between the metabolic state of cells and their “decision” to modify their lifestyle. Recently, transcriptional regulator ArgR has been shown to act as a key element in the arginine-dependent regulation of c-di-GMP turnover in *P. putida*. ArgR functions as a repressor of the arginine biosynthesis genes and as a positive regulator of arginine transport, thus altering the c-di-GMP levels. An *argR* mutant loses the response to arginine in terms of c-di-GMP accumulation and shows reduced expression of some biofilm matrix elements [44]. In turn, expression of *argR* is modulated by the second messenger, indicating the existence of a regulatory feedback loop (Figure 2).

From a biochemical point of view, it has been shown that the GGDEF-EAL-containing RmcA protein from PAO1 *P. aeruginosa* is able to recognise L-arginine through the periplasmic PBP domain (periplasmic binding domain) harboring the Venus flytrap fold [64]. This protein also contains a GGDEF-EAL tandem moiety whose final activity is a GTP-dependent phosphodiesterase [65]. Accordingly, the RmcA mutant accumulates more c-di-GMP than the wildtype PAO1 when L-arginine is the sole carbon source [64]. Recent data indicate that RmcA is involved in nutrient perception to sustain the maintenance of mature biofilm (rather than formation) once the attachment and the community is well-established; biofilms lacking RmcA or another phosphodiesterase named MorA cannot support nutrient limitation and finally die [66]. The name RmcA stands for the redox modulator of c-di-GMP [67]; this protein is involved in redox homeostasis (see below for more details); RmcA was found to inhibit colony wrinkling, a colony morphology which may improve the exchange of gases with the environment [67].

A similar strategy linking L-arginine sensing and c-di-GMP tuning has been observed in the DGC CdgH from *Vibrio cholerae* whose periplasmic PBP recognises L-arginine from the extracellular milieu, likely to regulate the intracellular levels of c-di-GMP via the cytoplasmic DGC domain [68].

Among the *Pseudomonas, P. aeruginosa* is an environmental organism able to infect a diverse range of life forms thanks to its metabolic versatility. In humans, *P. aeruginosa* is an opportunistic pathogen targeting immunocompromised and nosocomial patients [69]; moreover, this bacterium represents the leading cause of death in cystic fibrosis (CF) patients due to the chronic lung infections dominated by its biofilm [70].

The CF lung sputum is a nutrient-rich milieu able to support the high-density growth of *P. aeruginosa*, in part through the perception of amino acids to be assimilated and catabolised [71,72]. Multiple virulence-related phenotypes are modulated by CF lung components. In case of the environmental L-arginine, this amino acid may control the *P*. *aeruginosa* biofilm, which is relevant for infections, virulence [73,74,75] and antibiotic resistance [76]. A screening of the PA14 mutant library for mutants exhibiting an altered biofilm phenotype indicated that L-arginine metabolism is essential for *P. aeruginosa* biofilms [77]; the authors also found that the initial steps of pyrimidine biosynthesis were crucial, being necessary to sustain the pool of carbamoyl phosphate, a metabolite at the crossroads of both L-arginine anabolism and catabolism [77].

*P. aeruginosa* and other pathogens (both intracellular and extracellular) may alter the host L-arginine metabolism to sustain their homeostasis and mitigate the L-arginine-derived NO defense response [78]. In contrast, in the oral environment, L-arginine (and its catabolism) buffers the pH decrease due to the microbiota metabolism, thus protecting against the dental plaque biofilm [79]. L-arginine supplementation was also found to decrease the *P. aeruginosa* spread and sepsis in burn wound infections by influencing the virulence and may represent a promising phenotype-modulating tool for future therapeutic strategies [80].

Besides L-arginine, other polyamine precursors such as agmatine or exogenous putrescine upregulate biofilm formation in *Pseudomonas*, although the response to polyamines differs between the taxa [81]. Interestingly, the authors demonstrate that putrescine and L-arginine supplementation contributes to biofilm development (in terms of the extracellular structure) rather than to bacterial growth (cells) [81].

*P. aeruginosa* PA14 strain harbors an extra agmatine-induced *agu* operon responsible for agmatine utilization required for biofilm development, thus possibly providing further polyamines for maintenance or reparative processes or stabilization of DNA synthesis [82].

Given the relevance of L-arginine fermentation (i.e., the ADI pathway) in fueling ATP reservoirs under the anaerobic conditions that are frequently found at least in the innermost layers of the *P. aeruginosa* biofilm, this aspect was discussed in a dedicated paragraph.

## 5. Beyond Nutriment: L-arginine, Electrons and Oxygen

Oxygen restriction and microaerophilic environments are often found in the *P. aeruginosa* biofilm and, therefore, in chronic infection sites, such as the CF lung environment [83]. In particular, CF lungs promote the microaerophilic apparatus, including oxidase Cbb3-2, azurin and cytochrome c_551_ and the *Arc* operon encoding for the ADI pathway [83]. More generally, proteomic analysis carried out specifically in the anoxic portion of the biofilm displayed higher abundances of proteins from L-arginine and polyamine metabolism [84]. Chronic *P. aeruginosa* infections in mammals are regulated by L-arginine and the derived NO metabolism. Microaerophilic environments, such as in chronic infection sites, favour L-arginine fermentation, which in turn contributes to the NO deficiency typical of a reduced host defense response [85].

The metabolic reprogramming required to switch into the anaerobic metabolism contributes to the tolerance to antibiotics and resistance to specific molecules [76,83]. Indeed, the activation of ANR [86,87] and of the derived regulon, including L-arginine fermentation, represents a hallmark of the stress response leading to antibiotic tolerance and, more generally, is associated with chronic infections and persistence. 

Parallel to L-arginine fermentation, ANR also controls the long-term anaerobic survival, which involves pyruvate fermentation and the electron supply by phenazine redox cycling to obtain ATP [88,89]. Phenazines take advantage of the extracellular DNA to shuttle electrons within the biofilm structure to support anaerobic energy metabolism [90,91]. Briefly, a working phenazine oxidative cycle implies that in the deeper layers of the biofilm community, electrons accumulated for ATP production may be accepted by (oxidised) phenazines and delivered to oxidants (such as oxygen), which can be easily found in the outward part of the biofilm structure. Considering this, phenazines contribute to antibiotic tolerance together with the anaerobic stress response [92]. 

L-arginine was found to sustain both swarming inhibition and phenazine biosynthesis, although the molecular determinants linking phenazines and L-arginine are far from being elucidated [93]. When the phenazines’ electron-shuttling is poor due to mutations in the phenazines’ metabolism pathway or to robust biofilm formation, a wrinkling phenotype develops to increase the surface/volume ratio and, therefore, increase the gas exchange (i.e., oxygen availability). Therefore, the wrinkling phenotype controlled by RmcA (see above) in PA14 is related to energy production and redox balance (i.e., NADH/NAD^+^ ratio). Basically, it has been found that RmcA can be activated by oxidised phenazines to degrade c-di-GMP and inhibit colony wrinkling [67]. The authors hypothesised that electrons are perceived at the level of the PAS domain(s) located in the cytoplasmic portion of the protein. Given that other data showed that the periplasmic portion of PAO1 RmcA recognises and responds to L-arginine [64], it is not ruled out that *P. aeruginosa* checks the sustainability of its anaerobic ATP supply using this one-component system to finally reshape the colony morphology by tuning the c-di-GMP levels (Figure 3).

## 6. Concluding Remarks

Bacterial biofilms represent a biomedical and industrial issue, being related to chronic and nosocomial infections, antibiotic resistance, pipeline deterioration and goods contamination/degradation [94]. Although diverse, species belonging to *Pseudomonas* are relevant in all such issues due to their extraordinary metabolic versatility which confers high adaptability to these bacteria [95]. Nevertheless, the *Pseudomonas* biofilm may represent a biocontrol tool to protect against other microorganisms, including pathogens [96]. As robust, self-immobilised and self-regenerating biocatalysts, *Pseudomonas* biofilm-based bioreactors can be applied for waste treatment in biodegradation and bioremediation processes or used for fermentative biotransformation [97,98]. Particularly, *P. putida* is acknowledged as a possible cell factory through an engineering catalytic biofilm [99].

The environmental setting is crucial to sustain biofilm maintenance and dispersion, in particular nutrient availability [24]. In the *P. aeruginosa* biofilm, dispersal is inducible by nutrients, especially by increasing the amount of carbon sources [100]. In general, resource limitation and hostile environment promote biofilm formation in this bacterium [101].

It is now accepted that natural anti-biofilm agents, including metabolites, sensitise *P. aeruginosa* to antibiotics, and their use in combination with more traditional drugs is promising [102]. 

Among the most promising nutrients to tackle threatening human biofilms, L-arginine is one of the most versatile, being known to affect biofilm homeostasis and, simultaneously, augment the host’s defense by sustaining NO production.

Despite the large repertoire of literature data, the molecular mechanisms linking L-arginine perception and utilization to c-di-GMP and biofilms are starting to emerge [44,63,102]. Identifying the sensory domains of chemoreceptors specific for L-arginine and the regulatory elements involved in the connection between the amino acid and second messenger pools will provide new insights in biofilm adaptation strategies and the metabolic state of bacteria within a biofilm. 

Considering all this, the future dissection of the molecular mechanisms controlling the L-arginine-dependent metabolic reprogramming in *Pseudomonas* will be helpful for both biomedical and biotechnological applications.

## Figures and Tables

**Figure 1 ijms-23-04386-f001:**
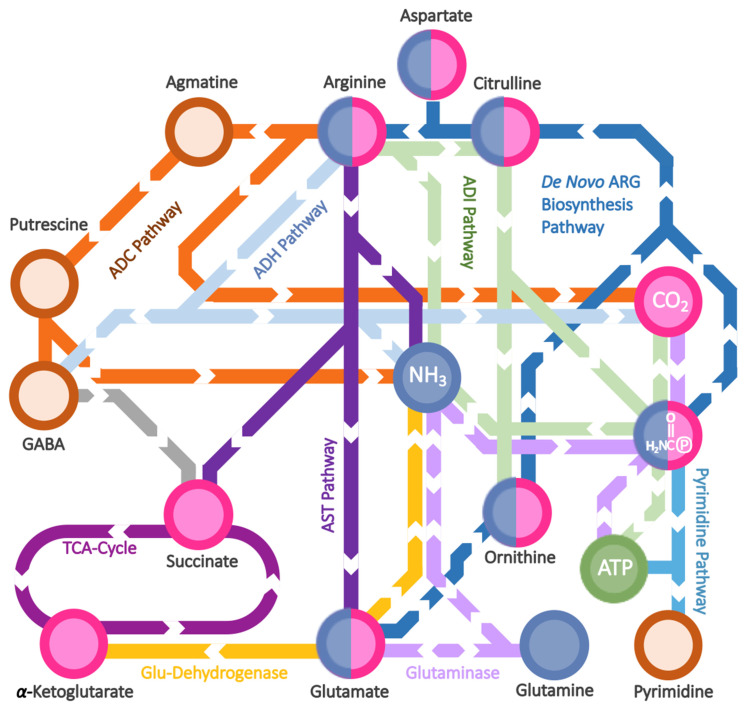
**L-Arginine metabolism in *Pseudomonas aeruginosa* in a metro-style map.** Nodes represent the most relevant metabolite(s) while different colors represent different metabolic categories. Each route represents the metabolic pathway named in the Figure with the same color, linking different nodes. The color code used for the nodes: (i) magenta and light navy nodes for C and N source metabolites, respectively; (ii) green for energy production; (iii) rust orange for other N metabolites. 2NHCO℗: carbamoyl phosphate.

**Figure 2 ijms-23-04386-f002:**
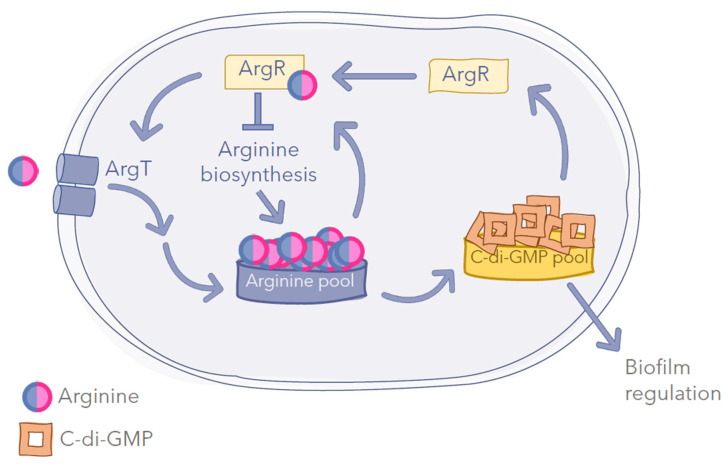
**Simplified model of the regulatory network connecting L-arginine metabolism, c-di-GMP signaling and biofilm formation in *P. putida* KT2440**. In the presence of arginine, ArgR positively controls arginine transport and negatively controls *de novo* arginine biosynthesis; intracellular arginine indirectly influences the c-di-GMP levels (and, therefore, biofilm formation) and, in turn, c-di-GMP modulates the expression of argR, thus establishing a feedback loop [63].

**Figure 3 ijms-23-04386-f003:**
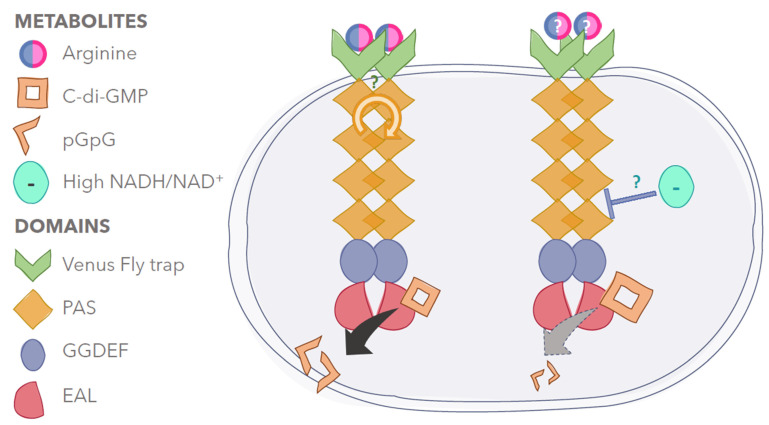
**Simplified model of the *P. aeruginosa* RmcA activation in response to environmental stimuli** such as arginine (on the left), which activates the phosphodiesterase activity [64], or electron availability (on the right), whose accumulation leads to reduced c-di-GMP hydrolysis [67]. The question marks indicate missing mechanistic details.

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
