# Peer review of "Nutrient Sensing and Biofilm Modulation: The Example of L-arginine in Pseudomonas"

_ijms, 2022, doi:10.3390/ijms23084386_

Round 1
Reviewer 1 Report
It is an interesting work that will be the basis of other research in the field.
That is why I recommend publishing the paper.
Please check how the years are written in references 17, 25, 27, 30...
Author Response
Response to Reviewer 1 in red.
It is an interesting work that will be the basis of other research in the field.
That is why I recommend publishing the paper.
Please check how the years are written in references 17, 25, 27, 30...
We wish to thank the reviewer for the positive evaluation. The query has been addressed.
Reviewer 2 Report
Comment
This paper provides a comprehensive description of Pseudomonas biofilms, L-arginine, and the current status of the induction and effects of L-arginine in Pseudomonas biofilms. Some comments are as follows:
- It is recommended that the purpose of this paper be made clear in the Abstract section!
- In the second part, "The importance of nutrients in the biofilm of Pseudomonas aeruginosa", it is suggested to write down which molecules play what role? It is suggested to expand this section considering these factors.
- It is suggested that Figure 1 can be depicted in more detail, which currently looks a bit t confusing.
- It is suggested to draw a diagram based on the content of the fourth part, briefly depicting this content.
- The description of the development status is mentioned in the abstract part of the article, but the description of this part is somewhat scattered and the description is little, and it is suggested to increase the description of this part in the article.
- It is suggested that the relationship between L-arginine, electrons and oxygen be represented in a diagram.
- In the last part it is suggested that there is too little content on the future outlook and it is suggested to add it.
- About one-third of the references in the article are from articles before 2010, and it is recommended to use more recently published articles.
In general, it is recommended to add charts for further analysis. Only one graph is not enough. The theme of the article is suggested to be further expressed, and it is also suggested to add expressions according to the comments.
Author Response
Response to Reviewer 2 in red.
This paper provides a comprehensive description of Pseudomonas biofilms, L-arginine, and the current status of the induction and effects of L-arginine in Pseudomonas biofilms. Some comments are as follows:
- It is recommended that the purpose of this paper be made clear in the Abstract section!
DONE.
- In the second part, "The importance of nutrients in the biofilm of Pseudomonas aeruginosa", it is suggested to write down which molecules play what role? It is suggested to expand this section considering these factors.
DONE.
- It is suggested that Figure 1 can be depicted in more detail, which currently looks a bit t confusing.
DONE. Being the Figure a metabolic metro-style map, we have improved the Figure legend.
- It is suggested to draw a diagram based on the content of the fourth part, briefly depicting this content.
DONE. A novel Figure 2 is now included.
- The description of the development status is mentioned in the abstract part of the article, but the description of this part is somewhat scattered and the description is little, and it is suggested to increase the description of this part in the article.
DONE. To avoid confusion a statement at the end of the first paragraph has been included.
- It is suggested that the relationship between L-arginine, electrons and oxygen be represented in a diagram.
DONE. A novel Figure 3 is now included.
- In the last part it is suggested that there is too little content on the future outlook and it is suggested to add it.
DONE. To avoid confusion we have clarified that this paragraph is only a concluding remarks section.
- About one-third of the references in the article are from articles before 2010, and it is recommended to use more recently published articles.
We agree with the referee, this is the reason why we strongly believe that this part of the nutrient sensing must be deeply investigated in the future.
In general, it is recommended to add charts for further analysis. Only one graph is not enough. The theme of the article is suggested to be further expressed, and it is also suggested to add expressions according to the comments.
DONE. We wish to thank the reviewer for the positive evaluation and fruitful suggestions.